# Quantity and Source of Protein during Complementary Feeding and Infant Growth: Evidence from a Population Facing Double Burden of Malnutrition

**DOI:** 10.3390/nu14193948

**Published:** 2022-09-23

**Authors:** Kulnipa Kittisakmontri, Julie Lanigan, Jonathan C. K. Wells, Suphara Manowong, Sujitra Kaewarree, Mary Fewtrell

**Affiliations:** 1Childhood Nutrition Research Centre, Department of Population, Policy and Practice, Research and Teaching Department, University College London Great Ormond Street Institute of Child Health, London WC1N 1EH, UK; 2Division of Nutrition, Department of Pediatrics, Faculty of Medicine, Chiang Mai University, Chiang Mai 50200, Thailand; 3Department of Pediatrics, Faculty of Medicine, Chiang Mai University, Chiang Mai 50200, Thailand

**Keywords:** protein intake, early-life nutrition, complementary feeding, animal source foods, double burden of malnutrition, infant growth, insulin, insulin-like growth factor-1

## Abstract

Background: While high protein intake during infancy may increase obesity risk, low qualities and quantities of protein contribute to undernutrition. This study aimed to investigate the impact of the amount and source of protein on infant growth during complementary feeding (CF) in a country where under- and overnutrition co-exist as the so-called the double burden of malnutrition. Methods: A multicenter, prospective cohort was conducted. Healthy term infants were enrolled with dietary and anthropometric assessments at 6, 9 and 12 months (M). Blood samples were collected at 12M for IGF-1, IGFBP-3 and insulin analyses. Results: A total of 145 infants were enrolled (49.7% female). Animal source foods (ASFs) were the main protein source and showed a positive, dose–response relationship with weight-for-age, weight-for-length and BMI z-scores after adjusting for potential confounders. However, dairy protein had a greater impact on those parameters than non-dairy ASFs, while plant-based protein had no effect. These findings were supported by higher levels of IGF-1, IGFBP-3 and insulin following a higher intake of dairy protein. None of the protein sources were associated with linear growth. Conclusions: This study showed the distinctive impact of different protein sources during CF on infant growth. A high intake of dairy protein, mainly from infant formula, had a greater impact on weight gain and growth-related hormones.

## 1. Introduction

The double burden of malnutrition (DBM)—the coexistence of under- and overnutrition—represents an emerging public health problem globally, especially for those in lower- and middle-income countries (LMICs) where eating habits are being transformed towards Westernized diets and lifestyles [1]. The World Health Organization (WHO) has emphasized the need for “double-duty actions” to prevent both forms of malnutrition [2]. While undernutrition and overweight were initially considered to affect different groups, they are increasingly recognized to occur within individuals through the life course^3^. This might potentially affect younger age groups, resulting in the combination of poor linear growth and overweight; however, evidence is lacking. This framework also recognizes that the two forms of malnutrition may have common risk factors in the form of unhealthy diets and environments [3,4]. Optimizing early-life nutrition by improving maternal nutrition, encouraging exclusive breastfeeding, and promoting appropriate complementary feeding (CF) are among the important actions identified to overcome the DBM [5].

The CF period is one of great change, when infants are introduced to foods other than milk [6] and protein intake increases; the percent of energy from protein (%PE) typically rises from around 5% to 15% when complementary foods become the major energy source for breastfed infants [7]. Protein is a key macronutrient promoting growth. However, to our knowledge, no studies have yet focused on the association between protein intake and growth in the context of the DBM. Previous studies have typically investigated the impact of protein intake in early life in populations facing either undernutrition or overnutrition [8], but not both.

In high-income settings, research has highlighted the association between “too much” dietary protein in early life and the increased risk of overweight/obesity in later childhood, while previous studies in resource-limited countries have focused on the association of “too little” high-quality protein with undernutrition and, particularly, stunting. There are no studies from LMICs investigating the effect of high protein intake in early life on the growth of infants and young children [8]. Furthermore, it is unclear whether all protein sources (i.e., dairy protein, non-dairy animal-based protein (ABP) and plant-based protein (PBP)) have the same effect on growth [9,10,11,12]. The most robust evidence, from a large, multi-center randomized controlled trial (RCT) in five European countries, reported that high protein intake from formula during infancy significantly increased weight gain, but not linear growth, in children aged 2 and 6 years [13,14].

Additionally, this RCT demonstrated that infants who received high-protein formula had significantly higher plasma insulin-like growth factor-1 (IGF-1) and urine C-peptide, an insulin derivative, at 6 months of age compared to those fed with low-protein formula or that were breastfed. However, the impact of other protein sources during CF was not investigated. During infancy, nutrition is the most important factor promoting growth via the GH–IGF axis [15], while amino acids derived from dietary protein are associated with IGF-1 and insulin secretion [16,17,18,19,20].

The aim of this study was to investigate associations of the amount and source of protein intake during the CF period with the growth of infants in Thailand, where the DBM is prevalent. Furthermore, plasma insulin, serum IGF-1 and insulin-like growth factor binding protein 3 (IGFBP-3) were also measured to support the clinical outcomes. We aimed to tackle a key question in the context of the DBM: how a specific component of infant diet may relate to markers of both undernutrition and overweight in early life.

## 2. Subjects and Methods

A multicenter, prospective, cohort study was conducted at three well-baby clinics in Chiang Mai, Thailand, between June 2018 and May 2019. Healthy term infants with birth weight ≥2500 g were recruited at age 4–6 months. Exclusion criteria were: infants with any underlying or chronic diseases; known cases of, or recovery from, protein-energy malnutrition; or those who regularly received medication except mineral and vitamin supplementation. Parents and legal guardians were provided with study information and gave written informed consent before enrollment. Ethics approval was obtained from the University College London Ethics committee, United Kingdom (Approval ID: 12551/001), and the Ethics Committee of the Faculty of Medicine, Chiang Mai University, Thailand (Approval ID: PED-2561-05287).

Data including demographics of infants, family characteristics, growth measurements and dietary assessments were collected at 6, 9 and 12 months (M) during routine child health surveillance clinic visits. Body weight and the recumbent length of infants were measured by trained health professionals using an electronic scale (TScale Electronics Mfg. Co., Ltd., Kunshan, Taiwan, precision ± 5 g) and a standard wooden measuring board (precision ± 0.1 cm). The weight-for-age (WAZ), weight-for-length (WLZ), BMI (BMIZ) and length-for-age (LAZ) z-scores (standard deviation scores) were calculated using WHO Anthro version 3.2.2 [21]. Stunting, wasting and underweight were defined as LAZ, WLZ or WAZ <−2 standard deviation score (SDS), respectively. Overweight and obesity were defined as WLZ more than +2 SDS [22]. The primary outcome was conditional growth at 12 months (see below).

Dietary intake was estimated using a food frequency questionnaire (FFQ) for the semi-quantitative estimation of habitual intake alongside a 24 h recall interview (24-HR) at all time points, and a 3-day food record (3-DFR) was also collected at 9 and 12 months for quantitative estimation. Initially, dietary data from the 3-DFRs were used to estimate the average energy and nutrient intakes at 9 and 12 months of age, while the 24-HRs were used to estimate those intakes at 6 months of age. However, in cases where the-3DFR was missing, dietary intake from the 24-HR was used instead. The FFQ was used to confirm the portion size if data from either the 3-DFR or 24-HR were unclear. Dietary intakes were converted to energy and nutrients using the Institute of Nutrition Mahidol University Calculation (INMUCAL)-Nutrient program version 4.0 (2018) developed by the Institute of Nutrition, Mahidol University, Bangkok, Thailand [23]. This programme provided information on total energy consumption (kcal/day), crude intakes of all macronutrients (g/day) and 8 micronutrients, as well as the caloric distribution from each macronutrient. In addition, the program also separately reported protein and iron intake from ASFs and plant-based foods (Appendix A).

Venous blood samples were obtained at 12 months of age. In total, approximately 2 mL of venous blood was obtained and kept at −20 °C until analyses were undertaken. Serum IGF-1 and IGFBP-3 were analyzed by a solid-phase, enzyme-labeled chemiluminescent immunometric assay using the IMMULITE^®^ 2000 system (Siemens Healthcare Diagnostics Products Inc., Devault, PA, USA). The intra- and inter-assay variation in these tests was less than 8%. Plasma insulin was analyzed by an electrochemiluminescence technique using the COBAS^®^ e411 analyzer (Roche Diagnostics Inc., Basel, Switzerland). The repeatability and intermediate precision of this technique were less than 5%.

Statistical analyses were performed using SPSS (IBM Corp. Released 2019. IBM SPSS Statistics for Windows, Version 26.0. Armonk, NY, USA: IBM Corp). The sample size calculation showed that at least 126 infants were needed to see differences of 0.5 z-score in WLZ at 12 months old between infants who regularly received red meat and those who did not [24]. For analyses, non-parametric data were natural log (Ln)-transformed prior to use in the regression models. Conditional growth status was calculated as z-score deviation from average size of the study population at 12 months of age, controlling for baseline size at 6 months. Simple linear regression was used to develop a formula predicting the average size of the study population at 12 months, while a positive and negative result indicated larger or smaller size than expected at follow-up, respectively, given their earlier size [25]. Demographic data, prevalence of malnutrition, CF practices and nutrient intake are described as means ± standard deviation (SD) and percentages depending on data characteristics. To investigate associations between protein intake and outcomes of interest, bivariate correlation and general linear models were performed. Pearson’s correlations were used to demonstrate relationships between the variables. Regression analysis was used to investigate the association between the main predictor (protein intake) and the primary outcome (conditional growth at 12 months old) and secondary outcomes, including insulin, IGF-1 and IGFBP-3. In order to investigate the effect of different protein sources, protein intakes were also divided into 3 groups: (1) milk protein—breast milk, formula, cow’s milk and other dairy products; (2) non-dairy ABP—meats, eggs and meat products; (3) PBP—cereals and legumes. Covariates in the regression models were selected using a directed acyclic graph (DAG). The DAG is considered as a statistical approach to identify confounding variables and reduce the risk of selection bias when estimating causality in observational studies. The DAG applied in this study was created using DAGitty.net version 3.0, 2020 [26]. To demonstrate the magnitude of effect, both correlation and regression coefficients were also reported with their 95% confidence intervals (CIs). Statistical significance was defined by a *p*-value less than 0.05.

## 3. Results

### 3.1. Demographic Data

One hundred and fifty healthy term infants were enrolled. There were four dropouts, and one infant was excluded due to developing a multiple food-protein allergy during the study period (Appendix A). Data from 145 infants (96.7%) were thus available for analysis. As shown in Table 1, there were almost equal numbers of male and female infants, while nearly two thirds of the infants were first-born. Mean parental age was around 30 years, and more than 50% attained at least a college degree. The majority of infants were living in extended, middle-class families where most families received a higher monthly income than the minimum wage in Thailand.

### 3.2. Prevalence of Malnutrition in the Study Population

At 12 months of age, the percentages of infants with wasting, underweight and stunting were 3.5, 4.1 and 4.8%, respectively, while only one infant (0.7%) was overweight. No infants in this cohort were classified as obese, or both wasted and stunted. According to the parental reports, over one-third of mothers and nearly two-thirds of fathers had overweight or obesity. Therefore, the prevalence of DBM at household level where underweight infants lived with parents who had overweight/obesity was 6.2% of all families.

### 3.3. Complementary Feeding Practices and Nutrient Intakes

Notably, 44.1% of infants were exclusively breastfed until 6 months of age, while 36.6% of all infants continued to receive only breast milk along with complementary foods until 12 months of age (Table 2). The mean age of introduction of CF was 5.7 ± 0.6 months. The most common first complementary food was rice with cooked egg yolk, while other non-dairy proteins such as meats and organ meats were introduced later. Mean protein intake during CF rapidly increased and reached its highest value at 12 months. In general, infants consumed more dietary protein than the Dietary Reference Intake for Thais 2020 (Thai DRI), as well as the intake recommended by the WHO. At 9 and 12 months of age, protein intakes were 2 to 3 times higher than the Thai and international recommendations (Figure 1). The average percentage of energy from protein (%PE) was 7.8, 12.6 and 15.6% at 6, 9 and 12 months of age, respectively.

Figure 1 shows mean protein intake (g/kg/day) of infants during complementary feeding. Compared to the recommendations suggested by the Thai dietary recommended intake (Thai DRI), the Institute of Medicine (IOM), the World Health Organization (WHO)/Food and Agriculture Organization (FAO)/United Nations International Children’s Emergency Fund (UNICEF) and the European Food Safety Authority (ESFA), the protein intake of this study population rapidly increased from 6 to 12 months and was higher than all recommendations at 9 and 12 months of age.

### 3.4. Association between Dietary Protein and Growth Outcomes

Infants were categorized into three groups based on the average %PE from 6 to 12 months of age; those in the highest and lowest quartiles had %PE ≥12.9% and ≤10.9%, respectively, while the median group received protein between these values. Infants in the highest quartile had significantly higher WAZ, WLZ and BMIZ at 12 months, while there was no significant difference in LAZ between groups (Table 3). Conditional weight-related z-scores (i.e., WAZ, WLZ and BMIZ) of infants in the high protein intake group were significantly higher compared to the median and low protein intake groups (Figure 2: 95%CIs shown in Appendix A), indicating that infants in the high protein intake group gained more weight than expected, given their baseline z-score at 6 months of age. However, there was no difference in the prevalence of all forms of malnutrition (i.e., underweight, wasting, stunting and overweight/obesity) between protein intake groups.

Figure 2 illustrates conditional growth status at 12 months. Infants who consumed protein in the highest quartile (black bar) had significantly higher conditional WAZ, WLZ and BMIZ compared to infants receiving protein in the median (dark grey bar) and lowest quartile (light grey bar). Although conditional LAZ was higher in the high protein intake group compared with other groups, there was no significant difference.

According to CF recommendations in Thailand [27] (Appendix A), infants should be given three main meals (i.e., breakfast, lunch, and dinner) from 9 months; thus, protein intakes during the early (6–9 months old) and later stages (9–12 months old) of CF were expected to be quite different. Therefore, average %PE during the early and later CF periods were separated for univariate analyses investigating the association of protein intake with conditional growth outcomes. The results in Table 4 indicate that protein intake from 9–12 months was significantly associated with conditional growth outcomes, whilst protein intake from 6–9 months of age was not. Thus, only protein intake from 9–12 months of age was included in the subsequent analyses.

According to the DAGs (Figure 3), the suggested covariates for the multiple linear regression model investigating the association of protein intake with linear growth were duration of predominant breastfeeding, type of milk feeding, non-protein energy intake at 6–12 months, maternal education, frequency of illness and family income. For ponderal growth including WAZ, WLZ and BMIZ, the DAG suggested duration of predominant breastfeeding, type of milk feeding, non-protein energy, maternal education, frequency of illness, maternal BMI and maternal age as covariates.

Figure 3 illustrates DAGs predicting conditional growth status (for either linear or ponderal growth) by protein intake during the CF period (main predictor). Black arrows indicate causal paths between the main predictors and outcomes. Dashed-grey arrows represent bias paths. Boxes with black frames show potential confounders.

All covariates suggested by the DAGs were included in the multiple regression models. There was no association between conditional LAZ and %PE from 9 to 12 months, or other covariates (Table 5). However, %PE from 9 to 12 months was associated with conditional WAZ, WLZ, and BMIZ (95%CI varied between 0.02 and 0.20). Considering different protein sources, only %PE from milk/dairy and non-dairy protein from 9 to 12 months were significantly associated with the weight-related parameters, while PBP was not (Table 5). Protein intake from milk had a stronger association with conditional weight-related parameters compared to other protein sources based on effect size (regression co-efficient (*β*)).

To differentiate the effect of milk protein from breast milk and that from dairy/infant formula on conditional growth outcomes, %PE from breast milk was subtracted from the %PE from formula, cow’s milk and other dairy products. The resulting variable was called “%PE from dairy vs. breast milk”. As shown in Figure 4, this variable was directly associated with weight-related parameters, suggesting that greater %PE from formula milk and dairy rather than breast milk was significantly associated with higher conditional WAZ, WLZ and BMIZ.

The findings suggest that a 1% increase in daily %PE from formula and dairy from 9 to 12 months of age was associated with a 0.18 (95%CI, 0.03, 0.32) and 0.16 (95%CI, 0.01, 0.30) standard deviation score (SDS) increase in conditional WAZ and WLZ, respectively, after adjusting for other protein sources, duration of predominant BF, non-protein energy consumption, type of milk feeding, maternal age, maternal education, maternal BMI, and frequency of illness.

A 1% increase in daily %PE from non-dairy ASFs from 9 to 12 months was also associated with a 0.10 (95%CI 0.02, 0.18), 0.12 (95%CI 0.04, 0.20) and 0.10 (95%CI 0.01, 0.18) SDS increase in conditional WAZ, WLZ and BMIZ, respectively, after adjusting for other protein sources, duration of predominant BF, non-protein energy consumption, type of milk feeding, maternal age, maternal education, maternal BMI and frequency of illness.

Scatter plots demonstrate associations between the percentage of protein energy from dairy sources (i.e., formula and cow’s milk), subtracted from the percentage of protein energy from breast milk (%PE dairy source vs. breast milk) and conditional growth at 12 months. The scatter plots show dose–response, positive associations between %PE dairy source vs. breast milk, and conditional WAZ, WLZ and BMIZ, but not conditional LAZ.

### 3.5. Association between Dietary Protein Intake and Blood Levels of IGF-1, IGFBP-3 and Insulin at 12 Months of Age

In order to investigate the association between dietary protein and weight-related growth parameters, IGF-1, IGFBP-3 and insulin were investigated at 12 months of age. Milk protein was the only food source that showed a significantly positive association with circulating IGF-1, IGFBP-3 and insulin (Table 6). However, a stronger association was found between “%PE from dairy vs. breast milk” and the IGF-1 level, suggesting the consumption of more %PE from formula and dairy than from breast milk was associated more strongly with IGF-1 level.

As shown in Figure 5, there were positive dose–response relationships of “%PE from dairy vs. breast milk” with all growth-related hormones after adjusting for sex. A 1% greater %PE from formula and dairy was associated with increasing blood concentrations of IGF-1, IGFBP-3 and insulin by 2.34 (95%CI 1.44, 3.23) ng/mL, 33.41 (95%CI 9.46, 57.37) ng/mL and 4 (95%CI 1, 7) %, respectively. Mean IGF-1, IGFBP-3 and insulin stratified by sex are given in Appendix A.

Scatter plots illustrate the associations between the percentage of protein energy from dairy sources (i.e., formula and cow’s milk) subtracted from the percentage of protein energy from breast milk (%PE dairy source vs. breast milk) and blood concentrations of IGF-1, IGFBP-3 and insulin at 12 months. The scatter plots show dose–response, positive associations between %PE dairy source vs. breast milk, and all laboratory markers after controlling of sex.

## 4. Discussion

This cohort study demonstrated that infants living in Chiang Mai, Thailand, consumed more dietary protein, mainly from ASFs, than Thai and WHO recommendations during the CF period. More importantly, the main results indicated that protein intake was significantly associated with weight-related parameters (i.e., WAZ, WLZ, and BMIZ) during the CF period after adjusting for potential confounders. Considering protein sources, the results showed a different impact of protein from diary and non-dairy ASFs. The predominant association with weight gain was from dairy protein—mainly formula and unfortified cow’s milk—whereas non-dairy ABP showed a lesser impact. Protein intake from formula and unfortified cow’s milk also showed positive associations with circulating IGF-1, IGFBP-3 and insulin at 12 months of age in a dose–response manner, independent of infant sex. There was no association of protein intake with linear growth markers, or of PBP with conditional growth outcomes in this cohort.

In contrast to a recent review highlighting that infants and young children in LMICs consumed less ABP compared with those from high-income settings [28], this cohort showed that infants living in northern Thailand consumed more dietary protein from ASFs than from plant-based foods during the CF period. It could be assumed that, in some LMICs, especially upper-middle-income countries such as Thailand, CF is now shifting towards a “Western style” diet. Recently, a cross-sectional study [29] and data from the national survey [30] in Thailand also reported that over 80% of protein in complementary foods came from ASFs. These findings are relevant to the current global situation in which many LMIC countries are transitioning to Western diets with high amounts of ASFs, even though this change may occur at very different rates across different countries [31,32].

Considering the relation between protein intake and growth outcomes, this study found that infants consuming protein in the highest quartile, with a median %PE of nearly 13%, had significantly higher weight-related z-scores at 12 months of age compared to those who had lower protein intakes. Interestingly, the median %PE of the high protein intake group was similar to a report based on European populations [33]. Michaelsen et al. [33] found that most studies in European countries showing a significant association between high protein intake and BMI at 12 months reported a %PE around 13%. Therefore, some experts agreed to recommend an upper limit of protein intake around 15%, with the aim of reducing the risk of childhood obesity in their populations [4,34,35]. In contrast, current international recommendations only recommend safe levels: lower limits of protein intake considered necessary to adequately support the normal growth of infants/children [36,37,38]. Given the dramatically increasing prevalence of overweight/obesity in young children in many LMICs, an upper limit of protein intake should be considered for international recommendations, and more studies in this specific context should be encouraged.

Furthermore, our findings also indicated dose–response associations between ABP and weight-related parameters regardless of the type of milk received or how much energy was provided from carbohydrate and fat, although the effects of dairy and non-dairy protein were different to some extent. When considering the concept of conditional growth [39], the outcomes can be interpreted as indicating that every 1% increase in %PE from either dairy or non-dairy ABP at 9–12 months of age is associated with a positive deviation in WAZ, WLZ and BMIZ from the expected values at 12 months, based on growth parameters at 6 months. Thus, these results suggest that higher protein intake from ASFs is associated with more rapid weight gain than the infant’s expected growth trajectory. Underpinning these clinical findings, our laboratory results showed that the higher consumption of dairy protein, mainly from formula and cow’s milk, significantly increased levels of circulating IGF-1, IGFBP-3 and insulin, which are the main hormonal regulators of human growth and may relate to increased adiposity [40,41]. A possible mechanism explaining the greater effect of dairy protein over other protein sources is the high proportion of leucine, a potent factor stimulating IGF-1 secretion in dairy protein compared to other food sources (14% vs. 8% of amino acids in dairy and meats, respectively) [42]. In addition, some evidence indicates that leucine also plays an essential role in the activation of the mammalian target of rapamycin (mTOR), which is the major regulator of growth and metabolism homeostasis in humans [43].

To our knowledge, this is the first evidence from an LMIC demonstrating an association between high protein intake and rapid weight gain, and the possible mechanism of this association through IGF-1, IGFBP-3 and insulin. More importantly, this cohort also showed the distinctive effect of different protein sources on infant growth, as previous evidence on this issue was inconclusive. The latest systematic review and meta-analysis examining the relationship between high protein intake and growth and risk of childhood overweight/obesity included no studies from LMICs [44]. This systematic review concluded that there is adequate evidence supporting a possibly causal effect of high protein, especially ABP, on BMI (dose–response effect), while limited evidence suggests that high protein intake may affect weight gain/weight-for-age score and the risk of childhood obesity. However, there were several inconclusive results, including the effect of high protein on linear growth and body composition [44]. The present study did not demonstrate a relationship between high protein intake and overweight/obesity due to the very small number of infants who were overweight/obese at 12 months old. However, there is evidence justifying the concern about the potential impact of high protein intake on overweight/obesity in this population. In 2019, the prevalence of overweight/obesity among Thai infants and young children aged less than 5 years rose to 12.7% [45] compared to the previous national surveys in 2009 and 2016 (8.5% and 8.2%, respectively) [46,47]. The daily protein intake reported in 2013 was similar to the present study; infants and young children aged 6 to less than 36 months had dietary protein intakes about 3 times higher than the Thai recommendations and nearly 80% of the protein was derived from ASFs, while total energy consumption was not different from the recommendation [30,48,49].

Notably, the literature from LMICs generally considers ABP as a preferred protein source due to its beneficial effect in preventing undernutrition [44,50,51]. Theoretically, protein from ASFs should provide adequate amounts of essential amino acids to meet the requirements of infants and children in order to prevent stunting [52]. A recent systematic review of studies on infants and children aged 6–60 months in LMICs did not find any significant associations between the consumption of ASFs and growth outcomes including weight, length/height and head circumference, though the included studies showed high heterogeneity [53].

The literature thus illustrates how ‘optimal’ protein intakes and sources during CF may differ in high-income and low-income settings. Reducing protein in complementary foods in European countries and the United States may help prevent childhood overweight/obesity, while promoting the consumption of ABPs in many low-income countries might mitigate the burden of wasting, stunting and micronutrient deficiencies. Nonetheless, for countries such as Thailand facing the DBM and nutritional transition, using either approach could be problematic. Therefore, such countries should ideally adopt recommendations related to dietary protein based on data from their population and avoid making assumptions by using dietary data from other countries.

More importantly, the distinctive effects of different protein sources should be taken into account when considering recommendations for dietary protein during the CF period. Current evidence suggests that dairy protein from formula and cow’s milk can promote rapid weight gain and could contribute to childhood obesity [54], while non-dairy ABP has a lesser impact on weight gain according to this cohort and other studies [55,56]. Therefore, to optimize protein intake during the CF period, nutritional policies focused on decreasing the intake of dairy protein, such as reducing the protein content in infant and follow-on formula and avoiding cow’s milk whilst encouraging mothers to continue breastfeeding throughout the first year of life, should be integrated into CF practices. In addition, non-dairy ABP enriched with essential micronutrients such as iron, zinc, iodine, and vitamin A should be promoted to provide adequate micronutrients whilst avoiding a high intake of dairy protein.

Finally, limitations of this study should be noted. First, the results from the present cohort cannot infer causality between dietary protein and rapid weight gain due to the observational study design. The association could be interpreted either way, and it is not possible to conclude whether dietary protein contributes to greater weight gain or whether parents of faster-growing infants provide more food, including protein. However, DAGs were applied to appropriately identify potential confounders and to avoid overadjustment and selection bias [57]. Second, the null effect of PBP on growth outcomes should be interpreted with caution because the PBP consumed by infants in this cohort was mainly cereals, whereas legumes and grains containing higher protein quantity and quality, which may be more frequently used in other LMICs, were rarely consumed. Third, the lack of a significant association between protein intake and linear growth may be due to lack of statistical power, as the sample size was calculated based on the expected difference in WLZ at 12 months between infants consuming ASF regularly and those who did not. Fourth, by assessing change in size between 6 and 12 months, and assessing complementary feeding during this period, some of the variability in growth that we quantified may have occurred prior to the dietary exposure. However, this makes any associations of growth and complementary feeding that we detect conservative. Lastly, it should be noted that “extra” weight gain from increasing intake of ABP cannot be assumed to indicate higher body fatness without additional evidence from body composition analysis, and we do not yet know how our findings will translate into the risk of overweight/obesity at later ages.

## 5. Conclusions

The present cohort provides evidence from a middle-income country that different protein sources may have contrasting influences on infant growth. While high protein intake from ASFs, especially formula and cow’s milk, during the CF period was associated with higher weight gain in a dose–response manner, the study did not find an effect on linear growth. Importantly, higher levels of IGF-1, IGFBP-3 and insulin in infants consuming higher amounts of protein from formula and cow’s milk provided mechanistic support for the clinical findings. However, further studies in populations facing the DBM and nutritional transition are needed to confirm the key findings from this cohort and to investigate the relationship between dietary protein and body composition. A longer follow-up period is also needed to see whether the study population consuming higher protein have a greater risk of overweight/obesity at later ages.

## Figures and Tables

**Figure 1 nutrients-14-03948-f001:**
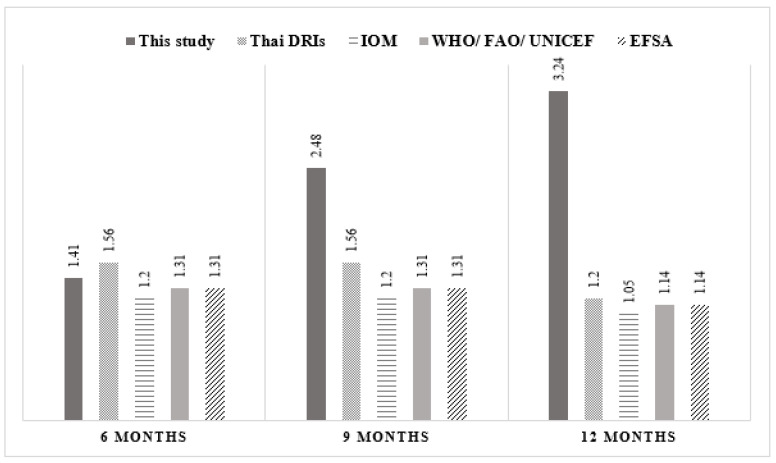
Comparison of protein intake (protein weight ratio) between this study and the Thai and international recommendations.

**Figure 2 nutrients-14-03948-f002:**
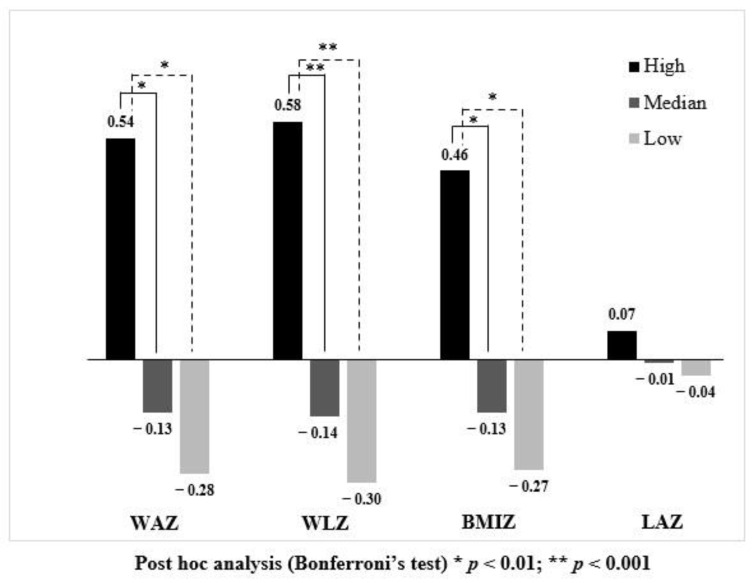
Comparison of conditional growth outcomes among infants consuming different protein intakes.

**Figure 3 nutrients-14-03948-f003:**
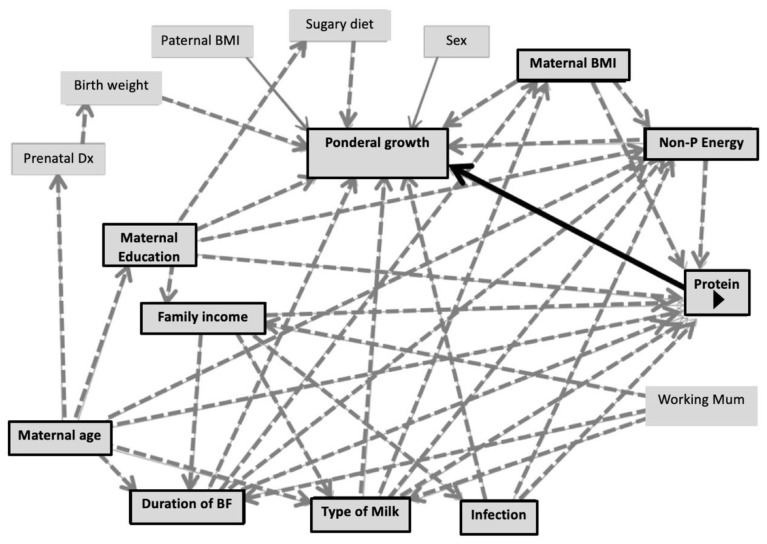
Directed acyclic graphs demonstrating co-variates of the association between protein intake and linear growth/ponderal growth.

**Figure 4 nutrients-14-03948-f004:**
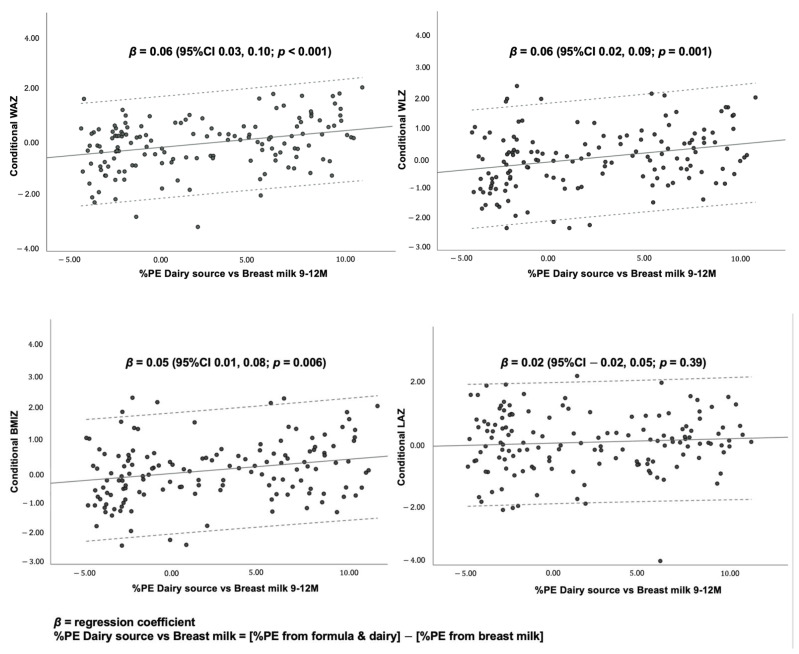
Scatter plots and linear regression statistics for conditional growth outcomes by %PE from dairy source vs. breast milk at 9–12 months.

**Figure 5 nutrients-14-03948-f005:**
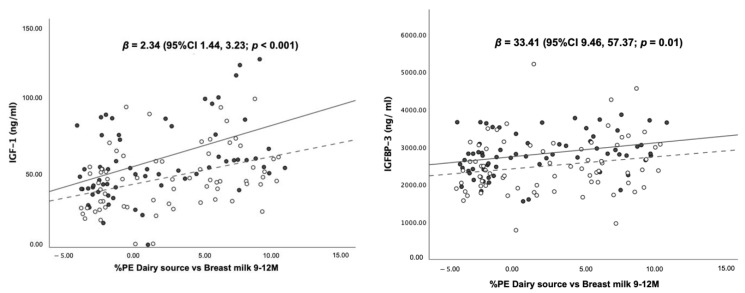
Scatter plots and regression statistics of blood levels of IGF-1, IGFBP-3 and insulin at 12 months of age by %PE from dairy vs. breast milk at 9–12 months (stratified by sex).

**Table 1 nutrients-14-03948-t001:** Demographic data and family characteristics (*n* = 145).

Demographic Data	Results
**Infants**	
Sex, female (n, %)	72 (49.7)
Gestational age, weeks (means ± SD)	38.8 ± 1.0
Route of delivery (n, %)	
- Vaginal delivery	96 (66.2%)
- Caesarean section	49 (33.8%)
Child order, first born (n, %)	93 (64.1%)
Birth anthropometry (means ± SD)	
- Body weight, kg	3.2 ± 0.4
- Length, cm	49.3 ± 1.9
- Head circumference, cm	33.3 ± 1.4
**Parents**	
Parental age, years old (means ± SD)	
- Mothers	29.8 ± 5.7
- Fathers	32.0 ± 5.9
Parental BMI, kg/m^2^ (means ± SD)	
- Mothers	22.8 ± 4.0
- Fathers	24.7 ± 3.6
Maternal educational attainment (n, %)	
- Did not receive formal education	2 (1.4)
- Below bachelor’s degree	74 (51.0)
- Bachelor’s degree and above	69 (47.6)
**Family characteristics**	**Results**
**Main caregivers** (*n*, %), choose more than 1	
- Mothers	134 (92.4)
- Fathers	6 (4.1)
- Grandparents	17 (11.7)
- Others	2 (1.4)
**Family type** (*n*, %)	
- Nuclear family	50 (34.5)
- Extended family	95 (65.5)
**Main financial providers** (*n*, %), choose more than 1	
- Mother	92 (63.5)
- Father	140 (96.6)
- Grandparents	13 (9.0)
- Others	2 (1.4)
**Family income per month** ^1, 2^, THB (*n*, %)	
- less than 10,000	11 (7.6)
- 10,000–29,999	65 (44.8)
- 30,000–49,999	51 (35.2)
- ≥50,000	18 (12.4)

N—number; SD—standard deviation. ^1^ Minimum wage in Chiang Mai was THB 320 per day during the study period. ^2^ Average monthly income of Thai families reported by the National Statistical Office of Thailand 2019 was THB 26,018.

**Table 2 nutrients-14-03948-t002:** Introduction of complementary feeding and milk feeding practices (*n* = 145).

Variable	Results
**Age of first introduction of complementary foods** (months), mean ± SD	5.7 ± 0.6
**Age of introduction of each food group** (months), mean ± SD	
- Rice	5.7 ± 0.6
- Fruits	5.8 ± 0.6
- Vegetables	5.9 ± 0.5
- Eggs	6.0 ± 0.5
- Meats	6.3 ± 0.9
- Dairy products (excluding infant/follow-on formula)	9.9 ± 2.2
**Breastfeeding practices**	
- Exclusive breastfeeding until 6 months of age, *n* (%)	64 (44.1%)
- Receiving only breast milk alongside complementary foods until 12 months of age, *n* (%)	53 (36.6%)
- Duration of exclusive breastfeeding (months), mean ± SD	4.4 ± 2.0
- Duration of predominant breastfeeding (months), mean ± SD	8.4 ± 4.4
**Formula and dairy products**	
- Receiving formula feeding, *n* (%)	87 (60.0)
- Receiving unfortified cow’s milk before 12 months of age, *n* (%)	21 (14.5)
- Duration of formula feeding (months), median (IQR)	3 (0, 9)

N—number; SD—standard deviation; IQR—interquartile range.

**Table 3 nutrients-14-03948-t003:** Comparison of growth among infants in different protein intake groups ^1^.

Growth Parameters	High (*n* = 36)	Median (*n* = 73)	Low (*n* = 36)	Mean Difference ^2^ (95%CI)		
				H vs. L ^3^	H vs. M ^3^	M vs. L ^3^
6M						
WAZ	−0.14	−0.40	−0.50	0.36 (−0.13, 0.85)	0.27 (−0.16, 0.69)	0.09 (−0.33, 0.52)
WLZ	0.02	−0.06	−0.05	0.07 (−0.46, 0.59)	0.08 (−0.37, 0.54)	−0.02 (−0.47, 0.44)
BMIZ	−0.08	−0.14	−0.16	0.08 (−0.46, 0.59)	0.06 (−0.40, 0.52)	0.02 (−0.44, 0.48)
LAZ	−0.15	−0.55	−0.66	0.50 (−0.01, 1.01)	0.40 (−0.05, 0.84)	0.11 (−0.34, 0.55)
9M						
WAZ	0.03	−0.46	−0.59	0.62 (0.16, 1.08 ^5^)	0.49 (0.09, 0.89 ^4^)	0.13 (−0.27, 0.53)
WLZ	0.14	−0.22	−0.24	0.38 (−0.10, 0.86)	0.36 (−0.06, 0.77)	0.02 (−0.39, 0.43)
BMIZ	0.09	−0.24	−0.26	0.34 (−0.14, 0.83)	0.32 (−0.10, 0.74)	0.02 (−0.40, 0.44)
LAZ	−0.17	−0.48	−0.69	0.52 (−0.01, 1.05)	0.32 (−0.14, 0.77)	0.20 (−0.24, 0.65)
12M						
WAZ	0.10	−0.45	−0.60	0.70 (0.24, 1.17 ^4^)	0.55 (0.15, 0.96 ^4^)	0.15 (−0.26, 0.56)
WLZ	0.25	−0.30	−0.39	0.64 (0.14, 1.16 ^4^)	0.55 (0.11, 0.99 ^4^)	0.10 (−0.35, 0.54)
BMIZ	0.29	−0.19	−0.31	0.60 (0.07, 1.13 ^5^)	0.48 (0.02, 0.94 ^5^)	0.12 (−0.34, 0.58)
LAZ	−0.19	−0.55	−0.64	0.45 (−0.07, 0.96)	0.35 (−0.09, 0.80)	0.10 (−0.35, 0.54)

CI—confidence interval. ^1^ Groups were classified by average percent protein energy (%PE) from all food sources at 6–12 months: high intake (H) infants received %PE in the highest quartile; median intake (M) infants received %PE in between the highest and lowest quartile; low intake (L) infants received %PE in the lowest quartile. ^2^ One-way ANOVA (eta-squared); ^3^ post hoc analysis (Bonferroni’s test); ^4^ *p* < 0.01; ^5^ *p* < 0.05.

**Table 4 nutrients-14-03948-t004:** Pearson’s correlations between protein intakes during two different periods (6–9 and 9–12 months of age) and conditional growth.

Conditional	Average %PE 6–9 M		Average %PE 9–12 M	
	r	*p*-Value	r	*p*-Value
WAZ	0.17	0.04	0.26	0.002
WLZ	0.16	0.06	0.23	0.006
BMIZ	0.12	0.16	0.20	0.02
LAZ	0.09	0.26	0.07	0.39

%PE—percent protein energy; r—correlation coefficient; WAZ—weight-for-age z-score; WLZ—weight-for-length z-score; BMIZ—body mass index z-score; LAZ—length-for-age z-score.

**Table 5 nutrients-14-03948-t005:** 1 Multiple linear regression analyses investigating associations between protein intake from all sources at 9–12 months and conditional growth.

Predictor and Co-Variates	Conditional WAZ		Conditional WLZ	
	*β*	95%CI	*β*	95%CI
%PE 9–12 M	0.11	0.03, 0.18 ^1^	0.12	0.05, 0.20 ^1^
Duration of predominant BF	0.02	−0.05, 0.08	0.02	−0.05, 0.09
Type of milk 9–12 M	0.10	−0.25, 0.45	0.19	−0.16, 0.55
Non-protein energy 6–9 M	0.002	0, 0.004	0.002	0, 0.004
Non-protein energy 9–12 M	<0.001	−0.001, 0.002	−0.001	−0.002, 0.001
Maternal education	0.06	−0.07, 0.18	0.05	−0.08, 0.17
Frequency of illness	−0.02	−0.16, 0.12	−0.03	−0.17, 0.12
Maternal BMI	−0.02	−0.06, 0.03	−0.01	−0.05, 0.03
Maternal age	0.001	−0.03, 0.03	0.001	−0.03, 0.03
**Predictor and** **Co-Variates**	**Conditional BMIZ**		**Conditional LAZ**	
	** *β* **	**95%CI**	** *β* **	**95%CI**
%PE 9–12 M	0.10	0.02, 0.18 ^2^	0.01	−0.07, 0.09
Duration of predominant BF	0.03	−0.04, 0.10	−0.02	−0.08, 0.05
Type of milk 9–12 M	0.23	−0.13, 0.59	−0.19	−0.56, 0.18
Non-protein energy 6–9 M	0.002	−0.01. 0.004	<0.001	−0.002, 0.003
Non-protein energy 9–12 M	<0.001	−0.002, 0.001	0.001	−0.001, 0.003
Maternal education	0.08	−0.05, 0.20	−0.05	−0.18, 0.09
Frequency of illness	−0.01	−0.15, 0.14	0.001	−0.15, 0.15
Maternal BMI	−0.01	−0.05, 0.03	N/A	N/A
Maternal age	0.01	−0.02, 0.04	N/A	N/A
Family income	N/A	N/A	0.12	−0.10. 0.34
2 Multiple linear regression analyses investigating associations between protein intakes from different food sources at age 9–12 months and conditional growth.				
**Predictor and** **Co-Variates**	**Conditional WAZ**		**Conditional WLZ**	
	** *β* **	**95%CI**	** *β* **	**95%CI**
%PE Milk/dairy	0.18	0.03, 0.32 ^2^	0.16	0.01, 0.30 ^2^
%PE Non-dairy ASFs	0.10	0.02, 0.18 ^2^	0.12	0.04, 0.20 ^1^
%PE Plant-based foods	0.15	−0.15, 0.45	0.16	−0.15, 0.46
Duration of predominant BF	0.02	−0.05, 0.09	0.02	−0.05, 0.09
Type of milk 9–12 M	−0.04	−0.46, 0.09	0.13	−0.30, 0.56
Non-protein energy 6–9 M	0.002	0, 0.004	0.002	0, 0.004
Non-protein energy 9–12 M	<0.001	−0.001, 0.002	−0.001	−0.002, 0.001
Maternal education	0.05	−0.07, 0.18	0.05	−0.08, 0.17
Frequency of illness	−0.02	−0.16, 0.12	−0.02	−0.16, 0.12
Maternal BMI	−0.01	−0.06, 0.03	−0.01	−0.05, 0.03
Maternal age	0.01	−0.03, 0.04	0.003	−0.03, 0.04
**Predictor and** **Co-Variates**	**Conditional BMIZ**		**Conditional LAZ**	
	** *β* **	**95%CI**	** *β* **	**95%CI**
%PE Milk/dairy	0.13	−0.02, 0.28	0.07	−0.08, 0.21
%PE Non-dairy ASFs	0.10	0.01, 0.18 ^2^	<0.001	−0.09, 0.09
%PE Plant-based foods	0.14	−0.16, 0.45	0.001	−0.31, 032
Duration of predominant BF	0.03	−0.04, 0.10	−0.01	−0.08, 0.06
Type of milk 9–12 M	0.17	−0.26, 0.60	−0.32	−0.76, 0.12
Non-protein energy 6–9 M	0.002	−0.01. 0.004	<0.001	−0.002, 0.003
Non-protein energy 9–12 M	<0.001	−0.002, 0.001	0.001	−0.001, 0.003
Maternal education	0.07	−0.06, 0.20	−0.05	−0.19, 0.08
Frequency of illness	−0.002	−0.15, 0.14	−0.002	−0.15, 0.14
Maternal BMI	−0.01	−0.05, 0.03	N/A	N/A
Maternal age	0.01	−0.02, 0.05	N/A	N/A
Family income	N/A	N/A	0.13	−0.10. 0.35

*β*—regression coefficient; CI—confidence interval; %PE—percent protein energy; BF—breastfeeding; WAZ—weight-for-age z-score; WLZ—weight-for-length z-score; BMIZ—body mass index z-score; LAZ—length-for-age z-score; N/A—not analyzed; ^1^ *p* < 0.01; ^2^ *p* < 0.05.

**Table 6 nutrients-14-03948-t006:** Pearson’s correlation between %PE at 9–12 months and blood levels of IGF-1, IGFBP-3 and insulin at 12 months of age.

Protein Intake (%PE) from	Correlation Coefficients (r)		
	IGF-1 (ng/mL)	IGFBP-3 (ng/mL)	Insulin (µU/mL)
All food sources	0.11	0.13	0.03
**Milk/Dairy** **%PE from dairy vs. breast milk**	**0.33** ^1^ **0.38** ^1^	**0.20** ^2^ **0.21** ^2^	**0.20** ^2^ **0.20** ^2^
Non-dairy ASFs	−0.16	−0.04	−0.14
Plant-based foods	−0.11	−0.09	−0.06

ASFs—animal source foods; %PE—percent protein energy; ^1^ *p* < 0.001; ^2^ *p* < 0.05.

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
