# Peer review of "Quantity and Source of Protein during Complementary Feeding and Infant Growth: Evidence from a Population Facing Double Burden of Malnutrition"

_nutrients, 2022, doi:10.3390/nu14193948_

Round 1
Reviewer 1 Report
This paper has good data and potential to make an important contribution. However, there are several places where clarification of methods is needed, and some key conceptual issues that should be addressed.
Minor wording/ clarification issues:
1. In the methods section: When describing “conditional” variables, these are measures that control for size at 6 mo, not growth.
2. Table 2 typo: diary products
3. In the methods section, it is stated that FFQ, 24 hr recall and 3 day food record are collected: It would be helpful to state, upfront, what each measure is used for in the analysis.
4. Table 2: formula: receiving formula : is this ever during the first 12 months?
5. Categorization of % PE during the CF period: Initially is this based on the average across 3 time periods?
6. How is % energy from breastmilk determined?
Conceptual/substantive issues:
1. Growth, feeding, and dietary intake are all time-varying and it is particularly important to capture the sequence of events, especially during infancy when changes are rapid. A better approach would be to use a longitudinal model which specifies outcomes (size) measured after exposures. Conditional measures capture growth over a 6-month period: it is inappropriate to model this as a function of intakes that are measured after some of that growth has already occurred.
2. A longitudinal, repeated measures analysis that captures feeding and size at each time point would be more appropriate to capture the sequence of feeding and growth.
3. The DAG should differentiate those factors which influence size at 6 months and then growth between 6 and 12 months.
4. Other aspects of the model specification don’t make sense: by including duration of predominant BF, and type of milk you are including an aspect of protein intake. How do you interpret % protein when controlling for type of milk, which is the source of protein?
5. How is non-protein energy represented? Also as percent of energy? Why does the model not adjust for total energy intake?
6. Ordinarily, models that partition the role of energy sources control for total energy and then use energy density of macronutrient of interest as the main exposure so that the results can be interpreted as an isocaloric (substitution model) vs an “addition” model (See extensive work by Willett and colleagues, esp Willette’s Nutrition Epidemiology textbook.
7. A higher % energy from protein means lower % energy from fat and/or carbs: How does the composition differ according to type of protein (e.g. when protein is lower, is % energy from fat higher? Or does carbohydrate content vary?
8. Multivariable models do not include child sex: why? Are relationships similar in males and females? The IGF results appear to show different slopes in males and females.
9. Interpretation: given the focus on food sources, how does micronutrient composition of the diet vary by protein source and how do you then rule out micronutrient effects?
10. More should be said in the discussion about the selected nature of this sample, which includes offspring of highly educated parents and where the dual burden is not an issue as indicated by the near absence of undernutrition.
11. The suggestion regarding enrichment with ABP with micronutrients is not supported by this study.
Author Response
Reviewer 1
Conceptual/substantive issues:
- Growth, feeding, and dietary intake are all time-varying and it is particularly important to capture the sequence of events, especially during infancy when changes are rapid. A better approach would be to use a longitudinal model which specifies outcomes (size) measured after exposures. Conditional measures capture growth over a 6-month period: it is inappropriate to model this as a function of intakes that are measured after some of that growth has already occurred.
Reply – The authors understand the reviewer’s concern, and this was an issue we carefully considered when planning our analyses. We expected body size at 6 months to be a confounding factor influencing both protein intake (main predictor) and body size at 12 months (main outcome), and our preliminary analyses showed this was indeed the case (please see a supporting document 1); hence it was necessary to adjust for size at 6 months in our analyses. We chose to use conditional growth measures which adjust for body size at 6 months without adding the measurements at 6 months to the regression models as separate variables. We prefer this approach as the effect size (regression coefficient) provides information on whether the size at 12 months is more or less than expected given the size at 6 months. However, as shown in the supporting document, the findings are unchanged when using size at 12 months as the dependent variable and including size at 6 months as an independent variable. To address the reviewer’s concern, we added this concern as our limitation (please see page 16, the first paragraph). By assessing change in size between 6 and 12 months, and assessing complementary feeding during this period, some of the variability in growth that we quantify may have occurred prior to the dietary exposure. However, this makes any associations of growth and complementary feeding that we detect conservative.
- 2. A longitudinal, repeated measures analysis that captures feeding and size at each time point would be more appropriate to capture the sequence of feeding and growth.
Reply – As explained above, we considered different approaches. However, in preliminary analysis, we found that complementary feeding was not well established among Thai infants before 9 months because parents still provided only 1 or 2 main meals of complementary foods following the Thai guideline (please see appendix in the manuscript) and used less variety of foods, especially animal-source foods. We think our final analysis plan best reflects the fact that our main focus was the impact of the amount and type of protein from complementary foods on growth.
- The DAG should differentiate those factors which influence size at 6 months and then growth between 6 and 12 months.
Reply – The authors understand the reviewer’s point, but it was difficult to decide which factors promoted growth at 6 months versus 6 to 12 months of age. Generally, all factors included in the DAGs have been reported in the literature as influencing factors of growth during infancy, although the effect size of each variable may vary at different ages.
- Other aspects of the model specification don’t make sense: by including duration of predominant BF, and type of milk you are including an aspect of protein intake. How do you interpret % protein when controlling for type of milk, which is the source of protein?
Reply – Both the type of milk and the duration of breastfeeding could be potential confounding factors when investigating the impact of protein intake from complementary foods on the growth outcomes, as they can influence both protein intake and growth. Although there is some ‘overlap’ in the information provided by type of milk and duration of breastfeeding, this would only be problematic in the model if there is collinearity, which was not the case (see diagnostics below). Including both variables in the model allows us to assess the independent contribution of each.
- How is non-protein energy represented? Also as percent of energy? Why does the model not adjust for total energy intake?
Reply – The authors would like to divide the answers into 3 parts as follows;
5.1) Non-protein energy reflects energy intake from carbohydrate and fat which are the main energy sources while dietary protein mainly provides amino acids for building body structures such as muscle and bone, as well as other functional proteins in normal condition. The non-protein energy is used to describe the balance between energy and protein which means it indicates the amount of energy that can be used as fuel for metabolism.
5.2) The percent energy represents the distribution of each macronutrient compared to total energy intake. It is a better way to describe the amount of protein intake regardless of infant body size. Generally, infants with higher body weight trend to consume more energy (kcal/ day) and protein (gram/day) than their peers with lower body weight. By using %PE, the bias of body size could be lessened because the protein intake is converted to a proportion based on individual intake.
5.3) As the %PE was a main predictor in multiple linear regression analyses, it could be inappropriate to control for total energy intake because this variable also includes energy from protein. Instead, we included non-protein energy as a controlled variable to see whether %PE was associated with growth outcomes independently from non-protein energy which is the actual energy source.
- 6. Ordinarily, models that partition the role of energy sources control for total energy and then use energy density of macronutrient of interest as the main exposure so that the results can be interpreted as an isocaloric (substitution model) vs an “addition” model (See extensive work by Willett and colleagues, esp Willette’s Nutrition Epidemiology textbook)
Reply – The authors are grateful for the reviewer’s advice. We appreciate that adjusting for energy using the methodology above might help interpretation of findings. However, we chose to conduct our analysis with protein expressed as a proportion of energy intake as all previous research on this topic had used this method. Adopting the same methodology allowed us to compare our findings with the work of others.
- A higher % energy from protein means lower % energy from fat and/or carbs: How does the composition differ according to type of protein (e.g., when protein is lower, is % energy from fat higher? Or does carbohydrate content vary?)
Reply – Results from a preliminary analysis did not show significant differences in average % energy from carbohydrate and fat among infants in the different protein intake groups at each time point. Please see the bar chart below.
Minor wording/ clarification issues:
- In the methods section: When describing “conditional” variables, these are measures that control for size at 6 mo, not growth.
Reply – Thank you for the suggestion, the author have revised this word as shown in the method section (page 3, paragraph 4)
“Conditional growth status was calculated as z-score deviation from average size of the study population at 12 months of age, controlling for baseline size at 6 months.”
- Table 2 typo: diary products
Reply – Thank you. The authors have corrected this in the revised manuscript.
- In the methods section, it is stated that FFQ, 24 hr recall and 3 day food record are collected: It would be helpful to state, upfront, what each measure is used for in the analysis.
Reply – Thank you for the suggestions. The authors added some sentences to clarify this point on page 3, paragraph 2.
“Initially, dietary data from the 3-DFR were used to estimate the average energy and nutrient intakes at 9 and 12 months of age while the 24-HR was used to estimate those intakes at 6 months of age. However, in cases where the-3DFR was missing, dietary intake from the 24-HR was used instead. The FFQ was used to confirm the portion size if data from either the 3-DFR or 24-HR were unclear.”
- Table 2: formula: receiving formula: is this ever during the first 12 months?
Reply – Yes, it is. Any formula being used at any time between 0-12 months of age was considered.
- Categorization of % PE during the CF period: Initially is this based on the average across 3 time periods?
Reply – Yes, it is. The authors revised the sentence to be clearer. Please see the result section 3.4 Association between dietary protein and growth outcomes.
“Infants were categorized into 3 groups based on the average %PE from 6 to 12 months of age…”
- How is % energy from breastmilk determined?
Reply – Regarding to the food composition software (the INMUCAL), a protein content of breast milk was reported based on volume of its daily intake thus, the authors could determine %PE from breast milk by calculating proportion of the energy from breast milk’s protein compared to a total energy intake that each infant had consumed each day.
Reviewer 2 Report
This study attempted to evaluate the impact of the protein type and quantity on growth in infants as they start complementary feedings. This was done using a prospective cohort study at three different centers. The patients all came from families with a socioeconomic status greater than the norm for their lower- and middle-income country. The authors have an underlying concern about infants being malnourished or overweight as the diet becomes more westernized with greater protein intakes.
Major Concerns:
1. Despite the Introduction focusing on under- and over nutrition, their cohort had very few infants in either category. If fact, their results show greater weight gain in infants that received more protein and specifically formula/dairy proteins but only one of the infants reached overweight/obesity levels. In the Discussion, the authors talk about the concerns associated with high protein intake being a precursor to obesity. They even state support for setting an upper limit for protein intake in young children. The issue is that they have not shown me enough supportive background evidence to raise my concern level. In fact, the authors did see a small number of undernourished infants. The logical first take would be that higher protein and higher weight might be a good thing. The authors need to present more evidence that support their concerns. I do appreciate that the last sentence of the Discussion correctly states the association of infant weight gain and later overweight/obesity is not known.
2. The authors did find increased weight gain specifically with higher protein intake and specifically formula/dairy proteins. However, they did not show significant improvement in linear growth. Length lags behind weight gain and is much harder to measure precisely. The issue here is probably not a big enough sample size to see a smaller effect in a relatively brief time frame is. The authors did a sample size calculation, and it was based on data concerning weight gain, but I suspect underestimated the numbers needed to show improved length. The authors need to address this potential statistical concern.
Minor Concerns:
1. The authors decided to correlate their anthropometric data with the biological markers of growth, insulin, IGF-I and IGFBP-3. However, it is not until the Discussion that they tell the reader what the relationship of these compounds are to growth. Not every reader has a background in growth factors and nutrition. It can be very brief in the Introduction or Methods but a statement of the relationship of the measured factors to childhood growth would be helpful.
2. Results section 3.3, second paragraph, the first sentence clearly is referring to figure “1” but does not say so.
3. In the Discussion, there is no discussion of any potential explanation as to why the formula/dairy proteins appeared to have a greater impact on weight gain. What is unique about these proteins or how they are delivered that could lead to a larger weight effect?

Author Response
Reviewer 2
Major Concerns:
- Despite the Introduction focusing on under- and over nutrition, their cohort had very few infants in either category. If fact, their results show greater weight gain in infants that received more protein and specifically formula/dairy proteins but only one of the infants reached overweight/obesity levels. In the Discussion, the authors talk about the concerns associated with high protein intake being a precursor to obesity. They even state support for setting an upper limit for protein intake in young children. The issue is that they have not shown me enough supportive background evidence to raise my concern level. In fact, the authors did see a small number of undernourished infants. The logical first take would be that higher protein and higher weight might be a good thing. The authors need to present more evidence that support their concerns. I do appreciate that the last sentence of the Discussion correctly states the association of infant weight gain and later overweight/obesity is not known.
Reply – The authors are grateful for the reviewer’s comment about this issue. We agree that it is an important point to be highlighted in the manuscript. The authors added some discussion on why we are concerned by high protein intake in our population on page 15, paragraph 2
“The present study did not demonstrate a relationship between high protein intake and overweight/ obesity due to the very small number of infants who were overweight/ obese at 12 months old. However, there is evidence justifying concern about the potential impact of high protein intake on overweight/ obesity in this population. In 2019, the prevalence of overweight/ obesity among Thai infants and young children aged less than 5 years rose to 12.7% [45] compared to the previous national surveys in 2009 and 2016 (8.5% and 8.2%, respectively) [46, 47]. The daily protein intake reported in 2013 was similar to the present study; infants and young children aged 6 to less than 36 months had dietary protein intakes about 3 times higher than the Thai recommendations and nearly 80% of the protein was derived from ASF while total energy consumption was not different from the recommendation [30, 48, 49].”
- The authors did find increased weight gain specifically with higher protein intake and specifically formula/dairy proteins. However, they did not show significant improvement in linear growth. Length lags behind weight gain and is much harder to measure precisely. The issue here is probably not a big enough sample size to see a smaller effect in a relatively brief time frame is. The authors did a sample size calculation, and it was based on data concerning weight gain, but I suspect underestimated the numbers needed to show improved length. The authors need to address this potential statistical concern.
Reply – The authors are grateful for the reviewer’s suggestions. The authors have a sentence mentioning this issue as the limitation on page 16, paragraph 1.
“Third, the lack of a significant association between protein intake and linear growth may be due to lack of statistical power as the sample size was calculated based on the expected difference in WLZ at 12 months between infants consuming ASF regularly and those who did not.”
Minor Concerns:
- The authors decided to correlate their anthropometric data with the biological markers of growth, insulin, IGF-I and IGFBP-3. However, it is not until the Discussion that they tell the reader what the relationship of these compounds are to growth. Not every reader has a background in growth factors and nutrition. It can be very brief in the Introduction or Methods but a statement of the relationship of the measured factors to childhood growth would be helpful.
Reply – Thank you for the suggestion. The authors added sentences mentioning the association between those measured laboratories and infant growth in the Introduction (please see page 2, paragraph 4)
“During infancy, nutrition is the most important factor promoting growth via the GH-IGF axis [15] while amino acids derived from dietary protein are associated with IGF-1 and insulin secretion [16-20].”
- Results section 3.3, second paragraph, the first sentence clearly is referring to figure “1” but does not say so.
Reply – Thank you for your suggestion. The authors have specified number of the figure in that sentence.
“Figure 1 shows mean protein intake (g/kg/day) of infants during complementary feeding.”
- In the Discussion, there is no discussion of any potential explanation as to why the formula/dairy proteins appeared to have a greater impact on weight gain. What is unique about these proteins or how they are delivered that could lead to a larger weight effect?
Reply – The authors agree with the reviewer’s suggestion therefore, we added some discussion about this point on page 14, the last paragraph.
“A possible mechanism explaining a greater effect of dairy protein over other protein sources is the high proportion of Leucine, a potent factor stimulating IGF-1 secretion, in dairy protein compared other food sources (14% vs 8% of amino acids in dairy and meats, respectively) [42]. In addition, some evidence indicates that leucine also plays an essential role in activation of the mammalian target of rapamycin (mTOR) which is the major regulator of growth and metabolism homeostasis in human [43].”
